# The Qualitative and Quantitative Study of Radiation Sources with a Model Configuration of the Electrode System

Victor V. Kuzenov and Sergei V. Ryzhkov *

Thermal Physics Department, Bauman Moscow State Technical University, Moscow 105005, Russia;
vik.kuzenov@gmail.com
* Correspondence: svryzhkov@bmstu.ru; Tel.: +7-4-(99)-263-6570

**Abstract:** This research is devoted to the calculation and theoretical analysis of physical processes in the powerful electric discharge sources of UV radiation and shock waves with required and controlled technical and physical characteristics. Based on the calculations, the processes of converting the initially stored electromagnetic energy into internal, kinetic, magnetic and radiation energy formed in the electro-discharge plasma sources of plasma formation were studied, and the interactions of discharged plasma and its radiation with matter in different aggregate states were also studied. All the main magneto-plasma dynamic and radiative parameters of plasma formation in the electric discharge sources of UV radiation and shock waves are obtained.

**Keywords:** hot temperature; mathematical modeling; plasma dynamic; high current emitting discharge





## 1. Introduction

Electro-discharge plasma radiation sources of high spectral brightness and shock waves (SW) [1,2] are an independent class of discharges carried out with a powerful (up to 10 GW) pulse (discharge time $t_i$ = 10–500 μs) discharge of the main energy storage device to the interelectrode interval of a special configuration (Figures 1 and 2). Electromagnetic energy supplied to the discharge under the condition of pulse currents of high amplitudes ($j_m$ = 10 kA–1 MA) is converted into the internal, kinetic and radiation energy of a formed plasma formation or flow.

Depending on the type of prevailing plasma heating mechanism, all known electric discharge sources of the specified range of parameters can be divided into two main types:

(1.) Electric discharge sources with an ohmic mechanism of plasma heating and a high amplitude and density current (up to 1 MA/cm$^2$), at which the conversion to internal energy is due to the joule mechanism of energy dissipation and is associated with the transfer of energy of the electrons accelerated by the external electric field, yielding heavy plasma particles as a result of elastic and inelastic collisions;

(2.) Electro-discharge sources with a plasma dynamic mechanism of plasma heating, operating on the effect of the shock wave braking of plasma formation (flow) previously accelerated by electromagnetic forces and the thermalization of directed kinetic energy of a heavy component.

Sources of radiation with ohmic heating, or otherwise high-current emitting discharges (HCED), include discharges in which the action of intrinsic electromagnetic forces is directed to the magnetic localization of plasma formation, formed in the interelectrode gap. At the same time, the role of the kinetic energy of the plasma in the energy balance of the discharge should be minimal, with the maximum efficient conversion of joule energy going directly into the internal energy of the plasma. It should be noted that the ohmic mechanism of plasma heating itself will be effective if the energy transfer rate from the electrons to the ions exceeds the electric field energy transfer rate to the electrons. This condition is

fulfilled when the discharge current density is limited to the value $j_{cr} \approx en_e(3kT/M/)^{1/2}$, of which, in characteristic cases, ($n_e = 10^{18}$–$10^{20}$ cm$^{-3}$, $T \approx 10$–100 eV) is $j_{cr} \leq 1$ MA/cm$^2$. In addition, the effective introduction of joule energy into the plasma of such discharges should be carried out under the conditions of limiting the thermal expansion of the heated plasma (i.e., limiting the characteristic transverse size $b$), since the highest values of the active resistance of the plasma are realized such that $R_p \approx L/[\pi b^2 \sigma(T)]$ (where $L$ is the length of the interelectrode gap and $\sigma$ is the conductivity of the plasma with a temperature $T$). In HCED, the transverse restriction of the plasma channel can be carried out not only by a magnetic field, but also by gas and solid walls.

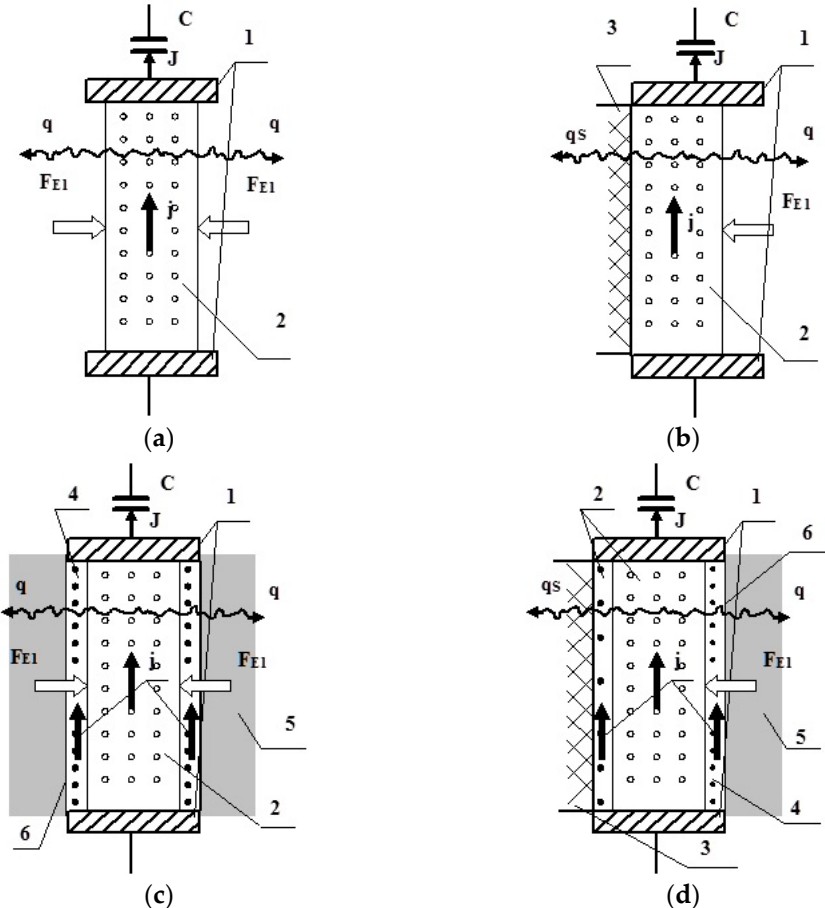

**Figure 1.** Circuits of open high-current emitting discharges (HCED). (**a**) Vacuum open unlimited discharge (OUD). (**b**) Vacuum open surface discharge (OSD). (**c**) Gas open unlimited discharge. (**d**) Gas open surface discharge. 1: discharge electrodes; 2: plasma of the gas or products of erosion of the structural elements of the electrode system; 3: dielectric solid surface; 4: shock-compressed gas area; 5: undisturbed gas medium; and 6: shock wave.

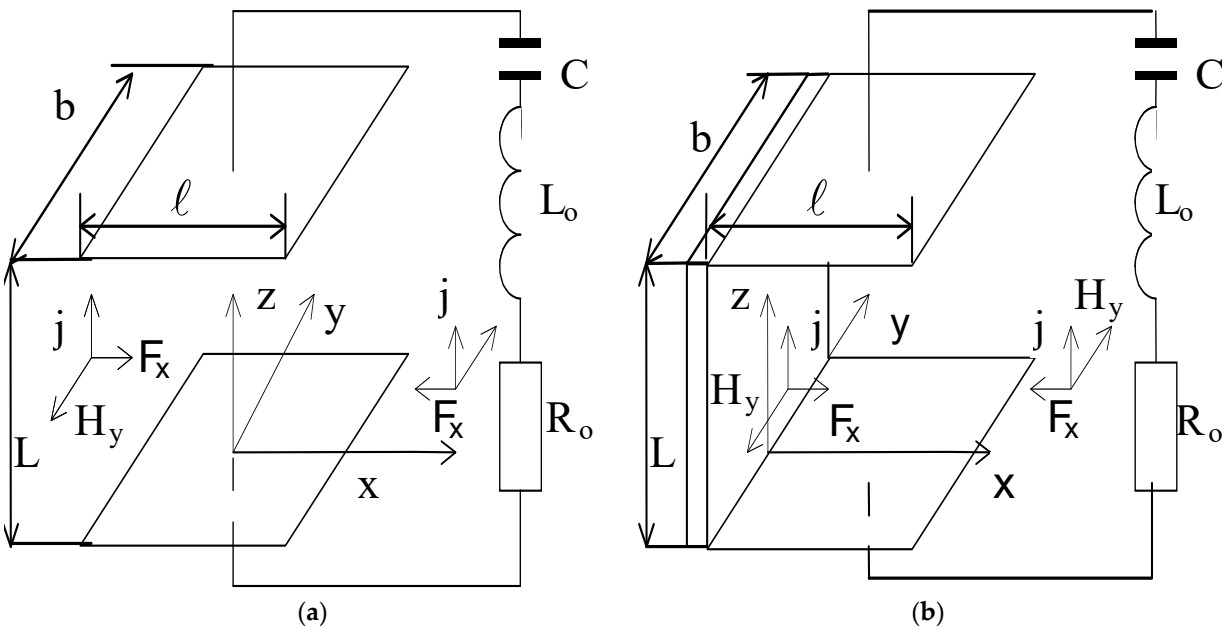

**Figure 2.** Model configurations of OUD (**a**) and OSD (**b**) electrode systems and the equivalent electrical circuits.

The most promising discharges, in terms of increasing the brightness temperatures of the HCED plasma, are the so-called open discharges developed in an unlimited (or partially limited) solid environment: open HCED in vacuums and gases. According to the current factors that limit the rapid expansion of the plasma channel, all open HCEDs are divided into the following categories:

(1.)　Vacuum open unlimited discharges (Figure 1a), with only compressive electromagnetic forces;

(2.)　Vacuum open surface discharges (Figure 1b), with compressive electromagnetic forces and a solid dielectric surface;

(3.)　Gas open unlimited discharges (OUD, Figure 1c), with the action of compressive electromagnetic forces and the back pressure of the surrounding gas discharge;

(4.)　Gas open surface discharges (OSD, Figure 1d), in which limitations of the thermal expansion of plasma are carried out simultaneously by electromagnetic forces, an external gas medium and a solid surface.

Model configurations of electrode systems and equivalent electric discharge circuits for OSD and OUD are shown in Figure 2. The electrode system represents two parallel flat strips of a certain width $b$ at a distance $L$ from each other. In the case of OUD, a flat surface is present in the interelectrode gap (IEG). The origin ($x = 0$) of the Cartesian coordinate system is located on the IEG surface for OSD and on the plane of symmetry of the discharge for OUD. The voltage supply from the capacitive accumulator to the electrodes provides the creation of electric field strength $E(x,t)$ in the interelectrode gap and current flow with a density $j(x,t)$ in the OZ axis direction. In the discharge region, an inherent magnetic field with an intensity $H(x,t)$ directed along the OY axis occurs. The configuration of the electrical conductors closing the electrical circuit is selected such that the resulting electromagnetic forces in the OUD are directed to the plane of symmetry of the discharge, and for the OSD, the electromagnetic forces are directed to one side from the IEG surface and to the side of the unperturbed gas medium (the IEG surface, providing general magnetic compression of the plasma).

Plasma formed in interelectrode space with time $t$, coordinate $x$ and values of density $\rho$, pressure $p$ and temperature $T$ propagates at speeds $u$ in the direction of the $X$ axis to the area of an initially fixed gas medium with known parameters.

The most promising direction for creating powerful sources of radiation with ohmic heating, meeting the requirements of many practical and scientific applications, is the use

of open surface HCEDs in gases (Figure 1d). The OSD group includes discharges whose plasma formations are formed above the cylindrical or flat surface of the non-destructible dielectric, which is not only a plasma-forming substance, but in some cases (for example, a linearly stabilized surface discharge (LSSD) in dense gases) performs the function of the necessary structural element responsible for initiating discharge.

One of the most important features of any radiation magnetoplasmadynamic (RMPD) process is the existence of strong nonlinear interactions between the various components of its processes: radiation, gas dynamic, electromagnetic and thermophysical. Due to the complexity of local diagnostics and experimental research of such RMPD processes and phenomena, in many cases, experimental data on the parameters and characteristics of the studied process or phenomenon are integral. It is often not enough to find causal relationships that determine the main physical patterns and identify effective ways to optimize and manage these processes. In such a situation, success in the creation of high-energy RMPD systems and installations for various purposes largely depends on the level of theoretical development. Moreover, the most effective method of research should be recognized as a computational experiment which, at some stages, allows for replacing expensive field experiments, and in a situation where experimental data are practically absent, numerical modeling remains the only way to obtain both qualitative and quantitative data for the process [3,4].

In general, the description of radiation magnetoplasmadynamic processes in the considered types of electric discharge sources should be carried out in a three-dimensional spatial approximation. The solution of the given system of equations in this case, even at present, seems very problematic. Therefore, it is advisable to consider and construct a hierarchy of spatially approximate models based on a complete system of equations, including the simplest one-dimensional models, which allows one to primarily identify the main qualitative features of the dynamics and spectral brightness characteristics of discharges.

## 2. Mathematical Models and Methods

This section is devoted to the development of mathematical models of RMPD processes in powerful electro-discharge sources of radiation of high spectral brightness and shock waves. Based on the analysis, it can be argued that the description of the RMPD processes taking place in them can be carried out in the framework of a single physical and mathematical model, formulated on the basis of the equations of multicomponent, single-temperature radiation magnetogasdynamics. Electromagnetic processes are described by a system of Maxwell and Ohm equations in plasma with finite conductivity. Radiation transfer is described within the multi-group diffusion approximation. The system is supplemented by equations describing the processes of heating and evaporation (within the framework of the model with the Knudsen layer) of the surfaces of the structural elements of the electrode assembly. The system is closed by the equations of the electric discharge circuit. This system of equations takes into account the methods for calculating the equations of the states of matter and the absorption coefficients of the natural radiation of plasma:

$$\partial\rho/\partial t + \partial(\rho u)/\partial x = 0,$$
$$\partial(\rho u)/\partial t + \partial\left(\rho u^2 + P\right)/\partial x = f_x, \ f_x = -\frac{1}{c}j_z H_y,$$
$$\partial(\rho E)/\partial t + \partial\left(\rho E u + P u + q_\Sigma\right)/\partial x = q_x,$$
$$q_x = j_z E_z, \ q_\Sigma = q_e + q_i + \sum_v q_v, \ P = P_e + P_i$$

(1)

where $t$ is time, $x$ is a coordinate, $\rho$ is the density, $u$ is the velocity along the $x$ coordinate, $P = P(\rho, \varepsilon)$ is the static pressure, $\varepsilon$ is the specific internal energy, $E = \left(\varepsilon + u^2/2\right)$ is the total energy of the flow, $q_v$ is the spectral radiation flux, $T_e$ and $T_i$ are the electron and ion temperatures, respectively, such that $T = T_e = T_i$, $f_x$ is the electromagnetic force, $q_x$ is the energy inflow from the electromagnetic field, $q_e = -\lambda_e grad T_e$, $q_i = -\lambda_i grad T_i$, $\lambda_e$ and $\lambda_i$ are the thermal conductivity coefficients of the electrons and ions, $j_z$ is the current density,

$\vec{H}(r)$ is the magnetic displacement vector, $P_e$ is the electron pressure, $P_i$ is the ion pressure and $v$ = (1, 2) for flat and axial symmetry.

The coefficients of electronic and ionic thermal conductivity $\lambda_{e,i}$ in the case of magnetized plasma, can be calculated using formulae [5]. The calculations of the parameters of the working media included in this system of equations of thermodynamic $e(T, \rho)$, $P(T, \rho)$ and optical $\chi_i(T, \rho)$ were carried out as part of the LTR approximation using the ASTEROID computer system developed by academician of the Russian Academy of Sciences S.T. Surzhikov [6], the Thomas–Fermi model with quantum and exchange amendments [7,8] and the average charge model [9].

The transfer of broadband radiation was taken into account by using a multi-group diffusion approximation, the equations of which are as follows [10]:

$$\frac{\partial q_v}{\partial x} + \chi_v c U_v = \chi_v 4\sigma T^4, \quad \frac{c}{3}\frac{\partial U_v}{\partial x} + \chi_v q_v = 0 \tag{2}$$

where $q_v$, $U_v$ are the spectral flux and bulk density of the broadband radiation, respectively, $c$ is the speed of light, $v$ is the number of frequency groups, $\chi_v$ is the spectral absorption coefficient, $n = 0$ is the flat layer, $n = 1$ is the infinite one-dimensional cylinder and $n = 2$ is the spherical symmetric case. Here, the value $q_v$ refers to the radiation flow in the direction of the $X$ axis. The value $\sigma_i$ was calculated using the following formula:

$$\sigma_i = 2k^2\pi[f(\alpha_{i+1}) - f(\alpha_i)](c^2 h^3), \alpha_i = \frac{hv_i}{kT},$$

$$f(\alpha) = \int\limits_0^\alpha \frac{\beta^3}{e^\beta - 1} d\beta \simeq \begin{cases} \alpha^3\left(\frac{1}{3} - \frac{\alpha}{8} + \frac{\alpha^2}{62.4}\right), & \alpha < 2 \\ 6.4939 - e^{-\alpha}\left(\alpha^3 + 3\alpha^2 + 6\alpha + 7.28\right), & \alpha \geq 2 \end{cases} \tag{3}$$

To solve the radiation transfer equations in a multi-group approximation, the spectrum was divided into groups, taking into account the peculiarities of the electronic structure of the atoms that were part of the plasma of the light-erosion vapors and ambient gas. The whole spectrum consisted of 7 groups, with interval boundaries [0.1–3.14–5.98–6.52–7.95–9.96–18.6–200] eV for the discharges in air and [0.1–3.14–5.98–11.62–15.76–18.82–27.63–200] eV for the discharges in Ar.

The magnetic induction equation for the field component is written as follows [10]:

$$\frac{\partial\left(H_y/\rho\right)}{\partial t} + \frac{1}{\mu}\frac{\partial\left(uH_y/\rho\right)}{\partial x} = \frac{c^2}{4\pi\mu\rho}\frac{\partial}{\partial x}\left(\frac{1}{\sigma}\frac{\partial H_y}{\partial x}\right),$$

$$j_z = \sigma\left(E_z + \frac{1}{c}uH_y\right) \tag{4}$$

Electrical conductivity was determined by the Spitzer formula [5], taking into account possible magnetization of the plasma.

The system of equations describing the processes of heating and evaporation of the IEG surface material and discharge electrodes under the action of incident and fully absorbed heat radiation flow with a density $q_x$, without taking into account the hydrodynamic processes in the condensed medium, consists of a quasi-uniform equation of thermal conductivity in a mobile (associated with the evaporation wave front) coordinate system with an $X$ axis perpendicular to the surface and a $Y$ axis parallel to the surface:

$$\frac{\partial T_s}{\partial t} = a_M\frac{\partial^2 T_s}{\partial z^2} + V_0\frac{\partial T_s}{\partial z} \tag{5}$$

with the following boundary and initial conditions: $k_m\frac{\partial T_s}{\partial z}(0, y, t) = q_z(0, y, t) - L_v\rho(0, y, t)v(0, y, t)$, $T_s(z\rightarrow\infty, y, t) = T_0$ and $T_s(0, y, t = 0)$

$= T_0$. The system of equations defining the kinetics of the condensed matter surface evaporation within the Knudsen layer model is as follows [11,12]:

$$\frac{T(0,\,t)}{T_s(0,\,t)} = \left[\sqrt{1 + \pi\left(\frac{(\gamma-1)m}{(\gamma+1)2}\right)^2} - \sqrt{\pi}\frac{\gamma-1}{\gamma+1}\frac{m}{2}\right]^2,$$

$$\frac{\rho(0,\,t)}{\rho_s(0,\,t)} = \sqrt{\frac{T_s(0,\,t)}{T(0,\,t)}}\left[\left(m^2 + \frac{1}{2}\right)e^{m^2}erfc(m) - \frac{m}{\sqrt{\pi}}\right] + \frac{1}{2}\frac{T_s(0,\,t)}{T(0,\,t)}\left[1 - \sqrt{\pi}me^{m^2}erfc(m)\right],$$

$$p_s(t) = p_1\exp\left[\frac{\Im L_v}{RT_1}\left(1 - \frac{T_1}{T_s(0,\,t)}\right)\right],$$

$$m = \frac{V(0,\,t)}{\sqrt{2RT(0,\,t)/\Im}},\quad \rho(0,\,y,\,t)v(0,\,y,\,t) = \rho_m V_0$$

(6)

where $T_s(x,t)$ is the condensed medium temperature at one point at a moment in time, $a_M, k_M, \rho_m$ are the temperature conductivity and thermal conductivity coefficients and the material density, respectively, $V_0$ is the evaporation wave rate, $p_s$ and $\rho_s$ are the condensed substance saturated steam pressure and density at surface temperature $T_s(x = 0, t)$, respectively, $R$ is the universal gas constant, $T_1$ is the condensed medium surface temperature value corresponding to the saturated steam pressure $p_1$, $L_v$ is the latent heat of evaporation, $\Im$ is the molar mass of steam, $T(0, t)$, $\rho(0, t)$ and $V(0, t)$ are the temperature, density and plasma velocity at the outer boundary of the Knudsen layer at a point ($x = 0$) and a time $t$, respectively, and $\gamma$ is the adiabatic index of condensed matter vapors.

The condition for applying a quasi-one-dimensional approximation was determined by the possibility of neglecting the term $\partial^2 T_s/\partial y^2$ in the two-dimensional thermal conductivity equation, which was justified by its smallness with respect to the derivative of the normal component (i.e., $(\partial^2 T_s/\partial y^2)/(\partial^2 T_s/\partial z^2) \approx 10^{-3} \div 10^{-4}$).

The electrical equation of the equivalent discharge circuit for the case when a capacitor battery with a capacitance $C$ is used as power energy storage has the following form [13]:

$$\frac{1}{c^2}\frac{dL_C J}{dt} + R_C J = U_k,\,\frac{dU_k}{dt} = -\frac{J}{C}$$

(7)

with the following initial conditions: $t = 0$, $J = 0$, $U_k(t = 0) = U_0$, where $J$ is the full discharge current, $U_0$, $U_k(t)$ are the initial and current voltages at the capacitor battery, $R_C = R_0 + R_p - \frac{1}{2c^2}\frac{dL_{n\pi}}{dt}$, $L_C = L_0 + L_p$, $L_0$, $R_0$ are the inductance and resistance of the external circuit and $L_p = \frac{c^2}{4\pi J^2}\int_V H^2 dV$, $R_p = \frac{1}{J^2}\int_V \frac{j^2}{\sigma}dV$, corresponding to the inductance and ohmic resistance of the discharge plasma, in which integration is carried out according to the volume of the calculated area, at the boundaries of which the magnetic field intensity value tends toward zero.

In the initial state, interelectrode space is uniformly filled with a gas medium with known properties and the initial thermodynamic parameters ($T_0 = 300$ K, $p_0 = 10^5$ Pa). This discharge step corresponds to the electrical breakdown of the gas gap between the electrodes and is modeled as an instantaneous occurrence at a certain time in the origin region ($x = 0$) of a narrow (thickness $\delta = 10^{-3}$ m) ionized gas layer with a temperature $T = 15$ kK.

The spectral flux and volumetric density of the broadband radiation at the initial moment of time are zero. At the moment of time $t = 0$, the intensity of the "seed" magnetic field $\vec{H}(r)$ in the environment is a fraction of a Tesla field.

In the system of electrical circuit equations with initial conditions $t = 0$, $J(0) = 0$, $U_C(0) = U_0$, the ohmic resistance and plasma inductance are determined by the following formulas:

$$R_p = L \left/ \psi b \int_0^l \sigma(x)dx\right., \quad L_p = (\ell n(2L/b + \psi x_T) + 0.5)L$$

where $x_T(t)$ is the coordinate of the current layer boundary in the discharge, $\psi = 1$ is the OSD and $\psi = 2$ is the OUD.

The values of the gas dynamic, radiative and electromagnetic parameters were set at the boundaries of the design area, which in these calculations was considered to be limited on the "left" side by the coordinate $x = 0$ (IEG surface is for the OSD, while the plane of symmetry is for the OUD) and on the "right" side by the undisturbed gas medium with the coordinate $x = \ell$.

The gas dynamic parameters of speed, density, pressure and internal energy at $x = 0$ for the OUD were determined by the conditions of "leakage." For the OSD, in the case of no evaporation, the "leakage" conditions were also used. In IEG evaporation, when $p_s(t) < p(0, t)$, the boundary values of the gas dynamic parameters were determined using Equations (1)–(4). From the "right side"—that is, at the point with the coordinate—for both bits, the conditions of "non-flowing" were set.

The following conditions were set at the boundaries of the design area to solve the radiation transfer with Equation (2): $\partial q_v(x = 0, t)/\partial x = 0, \partial q_v(x = \ell, t)/\partial x = 0$.

It should be remembered (when setting the boundary conditions on a magnetic field) that for the OSD, fundamentally different cases are possible that differ in the location of the external current supply buses. Here, an embodiment will be considered in which the magnetic field of the external current supply lines in the discharge zone is zero, and the boundary conditions on the magnetic field will take the following form:

$$H_y(0, t) = -\frac{2\pi J}{cb}, H_y(\ell, t) = -\frac{2\pi J}{cb} \text{ for OSD,}$$

$$H_y(0, t) = 0, H_y(\ell, t) = -\frac{2\pi J}{cb} \text{ for OUD.}$$

The computational algorithm, including the method of numerical solution of one-dimensional equations of the plasma dynamics, which is based on the method of fractional steps is described in detail in [9,14]. At the first fractional step, gas dynamic processes are taken into account (these processes correspond to the "hyperbolic" part of the considered system of equations). At the same time, the processes of radiation transfer, electromagnetic processes and processes of evaporation of the interelectrode insert, which take place in the electric discharge sources, are considered at the second fractional step.

For the "hyperbolic" part of the plasma dynamics equation system, a nonlinear, quasi-monotonic, compact polynomial difference circuit of an increased accuracy order [14] was used, and for the "parabolic" part of the equation system, a numerical method was used (which allowed for calculations in the cases of intense breaks in the transport coefficients) to solve the magnetic field diffusion equation and the thermal conductivity equation using a monotonized difference circuit of an increased accuracy order.

The method of calculating the transfer of broadband radiation was considered on the basis of multi-group diffusion approximation [9–11]. The time step necessary for integrating the above compact polynomial difference scheme was selected based on the condition that the Courant–Friedrichs–Lewy stability criterion was met. The numerical technique was verified by solving a number of test problems [9,14–18].

### 3. Results

In order to identify the main qualitative features of the dynamics and spectral luminance characteristics of the discharges at the main phase of joule energy extraction (i.e., in the first half-period of discharge current), it was possible to limit the scope one-dimensional approximation. For the OSD, the direction perpendicular to the surface of the interelectrode insert was highlighted as a priority, and for the OUD, the direction perpendicular to the axis of symmetry of the discharge was highlighted. In order to ensure the identity of the conditions for comparing RMPD processes in the OSD and OUD, the simplest geometry was chosen for both discharges: a flat discharge with a simple Z-pinch configuration.

Numerical studies of the plasma dynamic processes of discharges developing in an Ar or air medium (Air) at atmospheric pressure were carried out in variations with a length of

the interelectrode gap $L$ = 25–100 cm, level of value stored in the capacitive accumulator $C$ = 3–10 μF, energy $W_0 = CU_0^2/2 \approx 1 \div 100$ kJ and duration of the first half-period of discharge current $t_1$ = 5–10 μs. The range of change of the average specific (per unit length of the interelectrode gap) electric power released in the plasma was $P_e$ = 1–400 MW/cm. The width of the discharge electrodes was $b$ = 1 cm. As the IEG material for the OSD, corundum ($Al_2O_3$) ceramics were selected, the thermophysical characteristics of which were taken from [19].

At the initial $t = [0 - t_H]$, a substantially non-stationary stage of the discharge, a transition occurred from the initial ($t = 0$, $J = 0$) state (simulating the situation after the breakdown of the interelectrode discharge interval) to the quasi-stationary state. The results of the calculations showed that the value $t_H$ for the test conditions did not exceed the value of ~1 μs. During the rest of the time of the first half-period of the current ($t_H - t_1$), the phase of the main separation of joule power, the spatial distributions of the radiation plasma dynamic parameters of plasma formation were quasi-stationary in nature. With their appearance, one can judge the features of the emerging structures and dynamics of the propagation of discharge plasma and therefore talk about modes of discharge.

Both types of HCED (OSD and OUD) were characterized by strong discharge current attenuation with a maximum joule plasma energy release $W_1 = \int_0^{t_1} R_p J^2 dt \approx (0.4 \div 0.99)CU_0^2/2$ in the first half-period $t_1 \le 10$ μs of the discharge current $J$.

Based on the analysis of the degree of influence of individual discharge parameters, it was found that the main physical parameter determining the radiative and plasma dynamic characteristics was the average (for the first half-period) specific (per unit length $L$) rate of energy input into the discharge plasma $P_e = W_1/Lt_1$, the value of which in these calculations varied in the range of 1–400 MW/cm. In addition, it was clarified (Figure 3) that under the conditions of limiting the expansion of plasma formation by the gas medium and its own magnetic field, HCED, depending on $P_e$, could exist in three different quasi-stationary modes (explosive (EM), MHD and quasi-pitch (QP)) which differ in the structure and parameters of the plasma (see Figure 3).

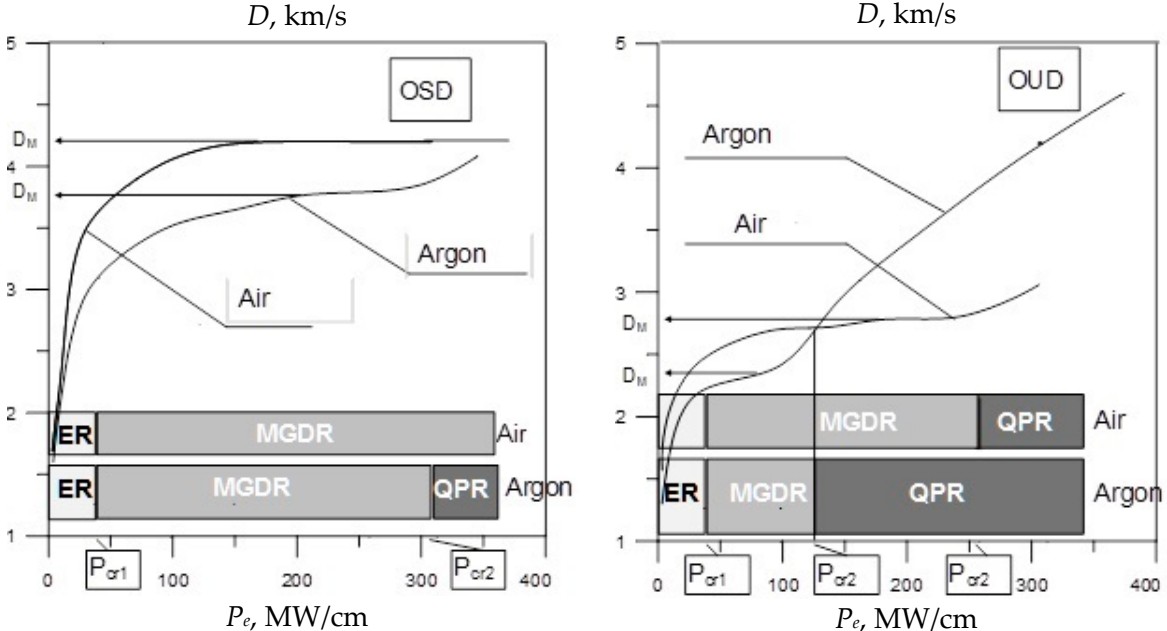

**Figure 3.** Mode diagram and propagation rate of the HCED outer boundary. ER: explosive regime; MGD: magnetogasdynamic mode; and QPR: quasi-pinch regime. Pcr1, Pcr2 are the critical values of the specific electric power, which are the lower and upper limits of the existence of the MHD mode.

Let us consider the peculiarities of behavior in the radiation plasma dynamic parameters of HCED for OUD and OSD. The boundary values of the average specific electric power, which determined the areas of existence of the modes, depended on the properties of the surrounding gas and were different for the OUD and OSD, reflecting the specific features of OSD associated with the presence of a solid dielectric surface, which was a rigid spatial limiter and a source of light-erosion plasma.

The calculations and their analysis showed that the HCED modes were associated with the plasma parameters (e.g., speed and density) in the boundary zones of the expanding plasma formation. In the case of $P_e < P_{cr1}$, an "explosive" mode was carried out, in which the external boundary of the discharge was a gas dynamic shock wave propagating into the surrounding gas at a speed of $D \sim P^{1/3}$. For the magnetogasdynamic mode ($P_{cr1} \leq P_e << P_{cr2}$), there was a limitation on the increase in velocity $D$ of the external discharge boundary for the increase of the specific electric power (Figure 3).

In the magnetogasdynamic mode, the effect of SW degradation appeared (Figure 4a), in which the compression ratio of the gas behind the outer break became smaller than the Gyugonio compression ratio and continued to decrease as the power embedded in the plasma increased. The effect of SW degradation was observed in experiments [20] under the conditions of strong external fields' action on the plasma of the spark discharge. In the quasi-pitch mode $P_e > P_{cr2}$, the external rupture turned (Figure 4b) from a compression rupture into a radiation magnetogasdynamic vacuum wave. The values $P_{cr2}$ were different for the OUD and OSD, reflecting the OSD's specific features associated with the presence of a solid dielectric surface, which was a rigid spatial limiter and a source of light-erosion plasma.

Here is a brief description of a magnetogasdynamic regime that is quite interesting from a practical point of view. As $P_e = W_1/Lt_1$ grows, the discharge current $J \approx Q_0/t_1 \approx (k_i P_e)^{1/2}$ and the discharge plasma temperature $T_2 \sim (t_1/c_{n\pi}\rho_0 b)P_{e\pi}^{1/3}$ increase. Magnetogasdynamic and radiation processes in the discharge plasma are intensified. An increase in the temperature $T_2$ causes an increase in the radiation flows $q_m \leq \sigma_{CB}T_2^4$ generated inside the plasma formation and shifts the maximum of the radiation spectrum beyond the "transparency window" of the gas surrounding the plasma.

Since the values of $P_e \approx P_{cr1} \approx 30$–$40$ MW/cm operate in the digit area's magnetic pressure $p_M = H_0^2/8\pi = \pi J^2/2c^2b^2 = k_m P_{e\pi}$, the size order is compared to the value of the gas kinetic pressure in the plasma of the category resulting from Joule heating of the plasma in the conditions of shock wave scattering such that $p_g \approx \rho_0(P_e/b\rho_0)^{2/3}$, $\beta = p_M/p_g \approx k_\beta P_e^{1/3} > 1$. The action of electromagnetic forces (along with the significant role of radiation effects) became a determining factor in the dynamics of the development of discharges and emerging structures. Therefore, this mode was called magnetohydrodynamic (MHD).

Spatial distributions of the OUD and OSD parameters in argon for $U_0 = 100$ kV, $L = 25$ cm and the $P_e \approx 100$ MW/cm variant are shown in Figure 4. The schematic spatial distributions were $(x)$ and $u(x)$. As can be seen from the above graphs, the discharge boundary in all cases was the compressed gas region I ($\rho_1/\rho_0 > 1$), and the density in the high temperature plasma zone II, in terms of the value, coincided with the density of the undisturbed gas. The maximum plasma temperatures of the discharges with a characteristic level of 60 kK were reached in the area of the OUD axis and approximately in the central part of the plasma region. In the direction of the outer discharge boundary, the temperature decreased monotonically, and the nature of this drop (especially in the area of the outer discharge boundary) depended on the properties of the surrounding gas.

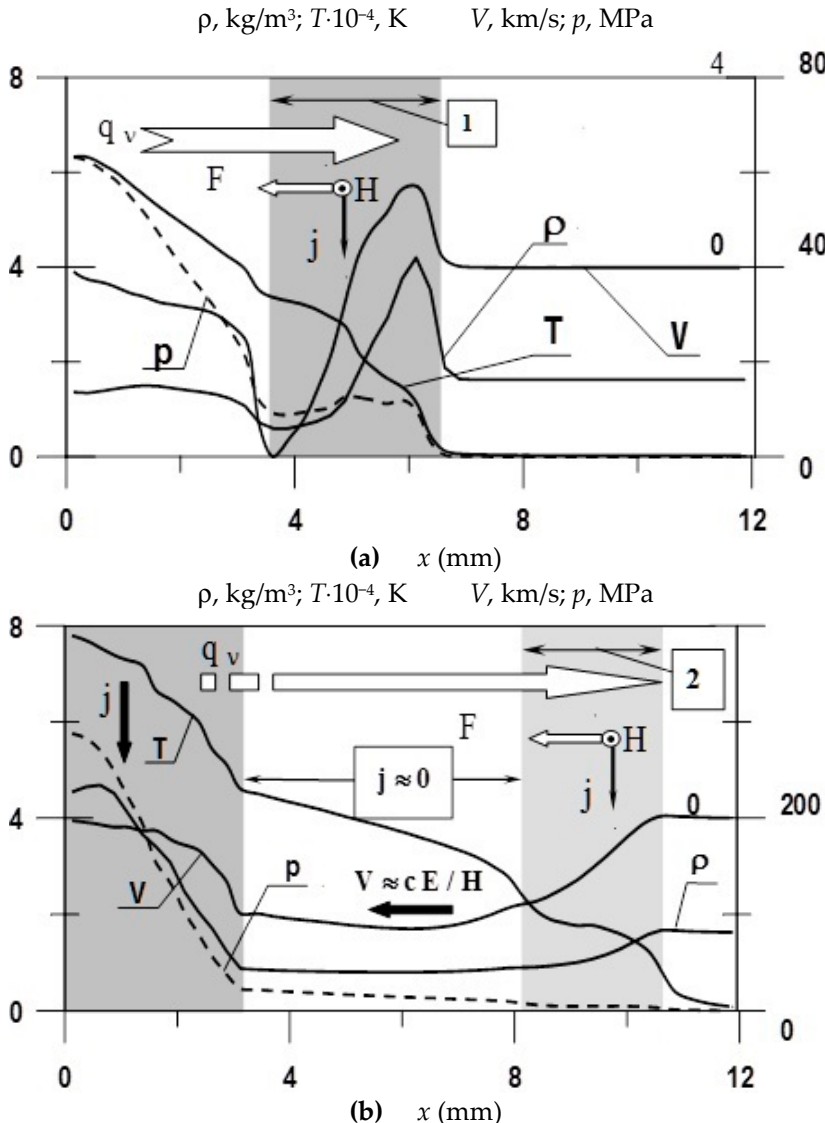

**Figure 4.** Effect of radiation magnetogasdynamic degradation of the shock wave in HCED, showing the distribution of gas dynamic parameters by spatial coordinate at the moment of maximum discharge current for the OUD in argon. 1: region of degrading MHD discontinuity; 2: region of the RMHD rarefaction wave. (**a**) $P_e \approx 100$ MW/cm (MHD mode); (**b**) $P_e \approx 300$ MW/cm (quasi-pinch regime).

For the OUD in argon, compressed layer I was sufficiently uniformly heated to temperatures of about 15 kK and to 5–10 kK in air. In the OSD, due to the cooling effect of light-erosion vapors, the temperature dropped in the direction of the surface. The action of the magnetic forces was clearly manifested in the presence of strong static pressure gradients in the plasma zone. The maximum pressures were achieved, as with the temperature, on the OUD axis and in the central part of the OSD plasma region. In the OUD, the action of the electromagnetic forces provides almost completed the inhibition of plasma in the area of the discharge axis and the occurrence of negative velocities in the zone adjacent to the GD rupture, clearly indicating the existing radiation magnetogasdynamic process of "outflow" of mass from the GD rupture region.

The following main features of the MHD mode can be noted.

The nature of the dependence of the GD fracture front speed on the specific electric power $P_e$ varied compared with the blasting mode. In the range $P_e \approx 30$–50 MW/cm, the growth rate $D$ from $P_e$ decreased, with a further increase from $P_e$ to $P_{cr2}$, while the speed slightly depended on $P_e$. The limit value of the velocity of the discharge boundaries into

the environment was reached, the value of which depended on the type of discharge and the properties of the environment.

The gas parameters behind the boundary of the GD break's front were not determined from the ratios for the simple strong gas dynamic SW. The gas in the GD fracturing region was heated to $T_1$ = 5–30 kK, and the compression ratio $\rho_1/\rho_0$ depended on $P_e$ and varied from $(\gamma+1)/(\gamma-1)$ at $P_e \approx P_{cr1}$ to $(\rho_1/\rho_0)_{\min} \approx 1$ at $P_e \approx P_{cr2}$.

High-temperature plasma zone II was magnetically compressed ($p_2 \approx p_M$) to a dense ($\rho_2 \geq \rho_0$) plasma region.

The qualitative picture of physical processes in bits for the MHD mode is as follows. Radiation from the high-temperature plasma region provides heating to temperatures of >20 kK for the GD rupture gas layers adjacent to the plasma zone. In these layers, current begins to flow, and its interaction with the magnetic field of the discharge current causes the formation of electromagnetic forces acting in the direction of the plasma zone. These forces inhibit the gas in these layers, which is equivalent to the "outflow" of gas from the GD fracturing region and corresponds to its entry into the high-temperature region.

Thus, the gas parameters in the GD rupture zone are determined by a complex of active radiation magnetogasdynamic processes:

(1.) Propagation of the front edge of the GD rupture into the undisturbed gas medium at a speed $D$ provides the shock wave seal of the gas under the condition of its additional heating with discharge plasma radiation;

(2.) From the "inner" side of the GD rupture, a radiation magnetogasdynamic discharge wave propagates, ensuring the outflow of gas mass to the plasma region.

As a result of the interaction of SW and the radiation magnetogasdynamic wave of the discharge, the intensity of which depends on $P_e$, the gas dynamic break characteristic of the MHD mode is formed, the parameters of which are determined by the value $P_e$. Reduction of the compression ratio $\rho_1/\rho_0$ with $P_e$ growth can be interpreted as the effect of radiation magnetogasdynamic degradation of the SW.

It is important to note that this effect is manifested in conditions where the propagation rate of the leading edge of the rupture $D$ is practically independent of $P_e$ (Figure 4), reaching the limit value $D_M$, which is associated primarily with the manifestation of the action of electromagnetic forces. With the same $P_e$ value, the propagation rate of the OSD boundary is approximately twice as high as that of the OUD, taking into account the expansion of the OUD plasma volume in two directions (see Figure 4). With all other conditions being equal (e.g., discharge length and electrical circuit parameters), the influence of the environment on $D_M$ is manifested through the value of the effective adiabatic index $\gamma_2$ in the plasma discharge zone. Since $\gamma_2(T_2, \text{Ar}) > \gamma_2(T_2, \text{Air})$, the velocity of the external discharge boundary in the air exceeds its value in the argon.

The MHD mode occurs in some range of values of the average specific electric power ($P_{cr1} \dots P_{cr2}$). The criterion determining the end of the MHD mode is the complete degradation of the GD gap as an external bit boundary. Physical reasons for the discharges output from the MHD mode are related to amplification (with $P_e$ growth) of the radiation flows and electromagnetic forces operating in the discharge zone. At $P_e > P_{cr2}$, the value absorbed by the GD rupture region, the radiation flows should be sufficient at time $\leq t_1/2$ to warm up the entire boundary region of the GD rupture to the temperatures at which the current flows in it, and the resulting braking electromagnetic forces will exceed the dynamic gas pressure in the GD rupture, thereby causing the radiation magnetogasdynamic effect of the GD rupture region and transition disappearing.

The critical values of the specific electric power of $P_{cr2}$, which are the upper limits of the existence of the MHD mode, depend on the type of discharge and the properties of the environment. As the calculations showed, for the OUD in argon, $P_{cr2} \approx$ 130–140 MW/cm, and for the OSD in argon, $P_{cr2} \approx$ 300 MW/cm. In other words, $P_{cr2}$ for the OSD in argon was approximately twice the $P_{cr2}$ for the OUD in argon. The main reason for this difference in $P_{cr2}$ was the difference in the GD break speeds for the OUD and OSD. For the OSD, higher speeds $D_M$ and levels of gas velocity $\rho_1 D_M^2/2$ in the GD break were required for the

braking large values of the electromagnetic forces acting in the GD break area, therefore leading to higher values of $P_e$.

The optical properties of the surrounding gas have a significant effect on the $P_{cr2}$ value. A necessary condition determining the possibility of complete degradation of the GD rupture is the condition of penetration of radiation, generated by the high-temperature plasma region of the discharge into the gas layer of the GD rupture with intensity and spectral properties, ensuring heating of this layer to certain temperatures ($T_1 \approx 20$ kK) at which the braking electromagnetic force effectively begins to act. A characteristic optical property of the air medium is the relatively low value of the spectral boundary of the "transparency window" of cold ($T \leq 10$–15 kK) air ($h\nu \approx 6$ eV). The increase in $P_e$ leads to an increase in the temperature of the plasma zone $T_2$ and a shift in the maximum of the radiation spectrum in the short wave range ($h\nu_M \approx 3kT_2$). With the temperature values $T_2 > 40$ kK of the plasma discharge zone being characteristic of the MHD mode ($P_e > 40$ MW/cm$^2$), the main part of the thermal radiation generated by the high-temperature plasma is in the region of short-wave radiation with quantum energy $h\nu_M > 12$ eV and, as can be seen, is outside the "transparency window" of cold ($T \leq 10$–15 kK) air. Radiation cannot penetrate the GD rupture, heat it to temperature $T_1 \approx 20$ kK or, during the first half-period $t_1$, provide the conditions necessary for the complete degradation of the GD rupture.

The magnetic Reynolds number $\text{Re}_m = VL_X/\nu_m$ varied in the range $\text{Re}_m \in [0.01, 1]$. Here, $L_X \sim 0.01$ m was the characteristic spatial dimension, and $\nu_m = c^2/4\pi\sigma \sim 3 \times (10^2-10^1)$ m$^2$/s was the magnetic viscosity of the plasma. This estimate shows the need to take into account the induced electric fields in LSSD modeling.

The interval of variation of the magnetic Euler number was estimated from the following relation: $\text{Eu}_m = P_M/(\rho V^2/2) \sim 10^2-10^{-1}$. Here, the value of the characteristic magnetic pressure $P_M = \frac{H^2}{8\pi}$ can be estimated using the expression $P_M \approx \pi J^2{}_X/(2c^2b^2) \approx (3-10^3)$ atm $= 3 \times (10^5-10^8)$ Pa, where $b \approx 0{,}01$ m is the characteristic transverse dimension and $J_X \approx (20-500)$ kA, being the characteristic value of the maximum value of the current flowing in the plasma, which can be approximately estimated using the expression $J_X \sim Q/(t_1/2)$ (where $Q = CU_O$ is the charge of the capacitor bank). Thus, it can be seen from this that the influence of electromagnetic forces on the motion of the discharge plasma can be significant.

The radiative Boltzmann number, defined as $Bo = \pi q b/P_e$ ($q$ is the characteristic total radiation flux LSSD) at brightness temperatures $T = (20-100)$ kK, had a level of values to the order of unity, which means that broadband thermal radiation of plasma plays an important role in LSSD energy balance, and radiation processes must be taken into account when developing a numerical model of a high-current-emitting discharge [21].

## 4. Discussion and Conclusions

It is important to note here that one of the interesting results [21] was the discovery of the anisotropy of the properties of the plasma and the gas dynamic boundary LSSD (in the directions of the $Z$ and $Y$ axes) at different values of the power energy parameters.

Therefore, to clarify the physical nature of the anisotropy of the properties of the LSSD plasma, additional work was performed, some of the results of which are presented in this article. In the presented article, a 1D mathematical model was used, in which one-dimensional radiation magnetogasdynamic equations were used in two spatially distinguished directions (one direction parallel to the surface of the interelectrode insert and one direction perpendicular to the surface of the interelectrode insert), qualitatively and correctly describing the LSSD plasma dynamics. In the 1D linearly stabilized discharge model, these two directions were deliberately separated from each other.

The purpose of the study using a 1D mathematical model was to obtain qualitative regularities of the plasma dynamic characteristics of the discharge (studying the effect of "degradation" of the gas dynamic parameters of the shock wave), which depend on the Joule energy release, electromagnetic forces and the radiation field and are not distorted by the mutual influence of the $Z$ and $Y$ directions.

The quantitative correctness of the results obtained using the 1D model of a linearly stabilized discharge was confirmed by comparing the 1D results and the results of the 2D numerical calculations. This comparison showed that there was good agreement between the results, indicating the validity of the 1D model of the linearly stabilized discharge adopted in the work (the difference in the results was at a 10–15% level).

Note that the time $t_2$ of the current rise in the LSSD was comparable to the characteristic time $\tau$ of the expansion of the plasma formation:

$$t2 \approx t_1/2 \sim \left[\frac{1}{J}\frac{dJ}{dt}\right]^{-1} \geq \tau.$$

Hence, it follows that the description of the physical processes in LSSD does not apply to any of the limiting cases ($t_1 >> \tau$, $t_1 << \tau$) when a simplified consideration of the plasma radiation dynamic processes would be possible.

Thus, in the LSSD, there is a strong mutual influence of the main physical factors (Joule input of energy into the discharge plasma, electromagnetic forces, thermal broadband radiation and evaporation of the material of the interelectrode insert). Therefore, these processes, which are of a complex nature, can be described only within the framework of the nonstationary mathematical model described in this article.

This paper presents a hierarchy of elements of RMPD mathematical models, which are designed to study the pulsed electric discharge sources of various classes. Data on plasma radiation dynamic processes and phenomena in pulsed erosion electric discharge sources of UV radiation and shock waves of various types were obtained with the help of the performed calculation studies. For these discharges, criteria were established that determined the discharge modes, differing in the degree of influence of electromagnetic processes and, as a result, the dynamics of formation and the spatial distributions of the plasma parameters. The effect of radiation magnetic degradation of the shock wave, manifested in a decrease in the compression ratio of gas at the front (compared to the Gyugonio adiabatic) and occurred in the conditions of radiation ionization of gas in the front area, and the action of braking electromagnetic forces was briefly described.

**Author Contributions:** Conceptualization, V.V.K. and S.V.R.; software, V.V.K.; validation, V.V.K.; writing—review and editing, S.V.R. All authors have read and agreed to the published version of the manuscript.

**Funding:** This research was partially funded by the Ministry of Science and Higher Education of the Russian Federation, grant number 0705-2020-0044.

**Institutional Review Board Statement:** Not applicable.

**Informed Consent Statement:** Not applicable.

**Data Availability Statement:** Not applicable.

**Conflicts of Interest:** The authors declare no conflict of interest.

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
