# Peer review of "The Qualitative and Quantitative Study of Radiation Sources with a Model Configuration of the Electrode System"

_symmetry, doi:10.3390/sym13060927_

Round 1

Reviewer 1 Report

The research is presented to the calculation and theoretical analysis of physical processes in powerful electric discharge sources of UV radiation and shock waves with the required and controlled technical and physical characteristics. Based on the calculations, the processes of converting  the initially stored electromagnetic energy into internal, kinetic, magnetic and radiation energy formed in electro-discharge sources of plasma formation were studied, the processes of  interaction of discharge plasma and its radiation with matter in different aggregate states were studied. All main magneto-plasma dynamic and radiative parameters of plasma formation in  electric discharge sources of UV radiation and shock waves are obtained. 

The research may be of interest to people involved in plasma research. 

I can recommend this article for publication as it is. 

Author Response

Authors thank Reviewers for a careful reading and helpful suggestions.

Reviewer 2 Report

A pdf file of comments is attached.

Author Response

(The authors gave the same response as above.)

Reviewer 3 Report

In this paper the authors presents a hierarchy element of radiation magnetoplasmadynamics mathematical model. The model is designed to study pulsed electric discharge sources of various classes. Data on radiation
plasma dynamic processes and phenomena in pulsed erosion electric discharge sources of UV radiation and shock waves of various types are obtained with the help of performed calculation studies. For these discharges, criteria are established that determine the discharge modes differing in the degree of influence of electromagnetic processes,
and, as a result, the dynamics of formation and spatial distributions of plasma parameters. 

The manuscript is highly technical and it can be interesting only to specific reader. I have no particular comments from the technical point of view.

My only suggestion is to improve the quality and discussion of the figures.

Author Response

(The authors gave the same response as above.)

Round 2

Reviewer 2 Report

The authors have responded my previous comments on the math model and the way of approach, showing how and why their simplification and assumptions are acceptable. I would like to suggest the authors add these explanation into either the manuscript or an appendix part. The refinements of pressure, temperature, time scales, viscosity of plasmas, Re and Eu number, etc. discussed in their response are very important supplemental information of their math model, although these details come from one of the previous publications as mentioned. However, it is still important to show the readers why using 1D in this paper is proper and sufficient. Therefore, please add these explanations.

Please correct some of the typos as I mentioned in my previous comments. Such as the extra "/" after "M" in the equation in Line 47 Page 2. Also, it will be better to replace all the "÷" sign with "-". Please double check the letter symbols, many of them are not italic. In Line 97 Page 4, the authors mentioned "Fig. 1g", but there is no subplot "g" in Fig. 1. Please correct it. 

The manuscript can be acceptable after a proper minor revision.

Author Response

Authors thank Reviewer 2 again and all the suggestions have been incorporated.
